# Bidirectional Associations between Parental Feeding Practices and Child Eating Behaviors in a Chinese Sample

**DOI:** 10.3390/nu16010044

**Published:** 2023-12-22

**Authors:** Jian Wang, Ruxing Wu, Xiaoxue Wei, Yan-Shing Chang, Xianqing Tang, Bingqian Zhu, Yang Cao, Yinghui Wu, Daqiao Zhu

**Affiliations:** 1School of Nursing, Shanghai Jiao Tong University, Shanghai 200025, China; jian.3.wang@kcl.ac.uk (J.W.); wu_ruxing@163.com (R.W.); weixiaoxue1998@163.com (X.W.); zhubq@shsmu.edu.cn (B.Z.); 2Florence Nightingale Faculty of Nursing, Midwifery and Palliative Care, King’s College London, London SE1 8WA, UK; yan-shing.chang@kcl.ac.uk; 3Department of Children’s Disease Prevention, Jinyang Community Health Service Center, Shanghai 200136, China; tangxianqing@126.com; 4Clinical Epidemiology and Biostatistics, School of Medical Sciences, Örebro University, 70182 Örebro, Sweden; yang.cao@oru.se; 5Unit of Integrative Epidemiology, Institute of Environmental Medicine, Karolinska Institutet, 17177 Stockholm, Sweden

**Keywords:** parents, preschool children, feeding practices, eating behaviors, bidirectional relationships

## Abstract

Background: Child eating behaviors (CEBs) and parental feeding practices (PFPs) play critical roles in childhood obesity. However, the bidirectional relationships between CEBs and PFPs remain equivocal. This longitudinal study aimed to explore their bidirectional relationships. Methods: A convenience sample of 870 parents with preschoolers was recruited in this longitudinal study (Shanghai, China). Three non-responsive feeding practices (NFPs), three responsive feeding practices (RFPs), five CEBs, and covariates were collected using validated questionnaires at baseline and the 6-month follow-up. Cross-lagged analyses using structural equation modeling (SEM) were performed to examine their bidirectional relationships. Results: Eight hundred and fifty-three parents completed questionnaires, with a response rate of 98%. The mean age of their children at baseline was 4.39 years (standard deviation = 0.72 years). Eighteen out of sixty longitudinal cross-lagged paths were statistically significant. Parental encouragement of healthy eating and content-restricted feeding were found to be bidirectionally associated with child food fussiness. Four parent-driven associations and one child-driven association were identified between RFPs and CEBs. For example, monitoring was negatively associated with children’s unhealthy eating habits (*β* = −0.066, standard error (SE) = 0.025, *p* < 0.01). Eight child-driven associations and one parent-driven association were observed between NFPs and CEBs. For example, higher child satiety responsiveness predicted a higher pressure to eat (*β* = 0.057, SE = 0.029, *p* < 0.01) and the use of food as a reward (*β* = 0.083, SE = 0.031, *p* < 0.01). Conclusions: There were bidirectional, parent-driven, and child-driven associations. Parents should be encouraged to adopt RFPs to shape CEBs. Increasing parents’ understanding of CEBs and providing them with reasonable coping strategies would help optimize PFPs.

## 1. Introduction

Childhood overweight and obesity have been linked to a series of health issues [1] and an increased likelihood of becoming obese as an adult [2]. Childhood overweight and obesity have become serious public health concerns, affecting 39 million children under the age of five globally in 2020 [3]. In 2020, about 7% of Chinese children under the age of six were overweight, and 3.6% were obese, accounting for the largest child population with obesity in the world [4].

Parental feeding practices (PFPs) play a significant role in childhood obesity [5,6]. Feeding practices mean specific practices that caregivers employ to manage what, when, and how much their children eat and shape their children’s eating patterns [7,8,9]. PFPs are classified into two food parenting constructs: non-responsive and responsive feeding [9,10]. Non-responsive feeding practices (NFPs) (also known as coercive control, self-centered feeding), such as the restriction of food and pressure to eat [7,8], have raised widespread concern due to their close relationships with obesity in children [5,11]. On the other hand, responsive feeding practices (RFPs) (e.g., modeling of appropriate food choices, monitoring, and encouragement of healthy eating) have been reported to result in a reduction in childhood overweight and obesity [6]. Additionally, PFPs may have a strong impact on child eating behaviors (CEBs) [12,13]. As a result, feeding practices have become focal points of family-based obesity prevention [14].

Cross-sectional studies have found significant associations between PFPs and CEBs [15,16,17]. For instance, one study (*n* = 977) conducted in Australia found that more overt parental restriction was correlated with increased food responsiveness in preschool children [17]. Theories of developmental psychology (e.g., Ecological System Theory) suggested bidirectional relationships between PFPs and CEBs [18,19]. A model proposed by Ventura and Birch (2008) [20] also indicated that parenting (e.g., parenting styles) and child eating (e.g., eating styles) are interrelated. Given the limitation of a cross-sectional design in determining causality, in recent years, researchers have highlighted the importance of using a longitudinal design to examine the relationships between PFPs and CEBs. Longitudinal studies have provided empirical support for the bidirectional relationships between PFPs and CEBs [21,22,23]. For instance, Jansen et al. [22] found that child food responsiveness at 3.3 years of age positively predicted the use of food as a reward one year later; conversely, the use of food as a reward for children at 4.3 and 5.3 years of age positively predicted the children’s food responsiveness at 5.3 and 6.3 years of age, respectively.

However, several studies revealed only unidirectional effects driven by child or parent behaviors [12,13,21,23]. Some studies showed that PFPs directly influenced CEBs (i.e., parent-driven association) [12,13]. For example, a cohort study (*n* = 797) conducted in Norway reported that greater parental use of food as a reward predicted more child emotional eating and food responsiveness after two years but found no significant predictive effect of CEBs on PFPs [12]. On the other hand, a few studies reported a child-driven association in which CEBs predicted PFPs. For instance, Jansen et al. [21] found that higher child food responsiveness at 4 years of age predicted more parental use of food as a reward at 9 years of age, while the use of food as a reward at 4 years of age did not prospectively predict child food responsiveness at 9 years of age. However, Lumeng et al. [24] reported no significant prospective relationships between parental pressure to eat and child food fussiness. In general, the current findings on the relationships between PFPs and CEBs based on longitudinal studies are inconsistent.

Most of the findings above were from studies conducted in developed countries. The relationships between PFPs and CEBs may change in developing countries because of their different feeding and belief cultures. The extent of adoption of specific feeding practices (e.g., use of food as a reward and modeling of healthy food choices) varied in different countries [15,25]. In China, parents prefer chubby children and usually do not view children with overweight or obesity as having a health problem [26]. Instead, most of them believe that higher weight represents better health and nutrition status. As a result, they often overfeed their young children or give children their favorite foods (e.g., energy-dense foods) to encourage them to eat more [27]. Recently, a cross-sectional study in China (*n* = 912) reported that greater restriction of food was associated with fewer unhealthy eating habits and lower food responsiveness in children [15], contrary to the findings from developed countries [28]. Collectively, PFPs often carry cultural and regional variations, which may moderate the relationships between PFPs and CEBs.

Overall, the findings about the bidirectional relationships between PFPs and CEBs have been equivocal. We aimed to examine the temporal directionality between a range of PFPs and CEBs in a Chinese sample with a prospective design. We applied a cross-lagged modeling approach to test their bidirectional associations while controlling for potential covariates [9]. To our knowledge, this is the first prospective study to explore their bidirectionality in China. The findings from this study will enhance our understanding of their relationships and inform future interventions to optimize parent–child interactions between PFPs and CEBs.

## 2. Methods

### 2.1. Study Design and Participants

A longitudinal study was conducted between October 2020 and July 2021 in Pudong District, Shanghai, China. We collected data at two time points (baseline T1 and follow-up T2) with a 6-month interval. This duration of follow-up was used to avoid missing critical periods of changes in relationships [9,13,24] and maintain a high retention of the follow-up sample.

We applied convenience sampling to recruit parents who were responsible for their preschoolers’ eating in the family from seven public kindergartens located in districts with various economic levels. We used the formula below to calculate the required minimum sample size, which was 467 [29]:$$N_{\varepsilon }=\frac{\delta_{1-\beta }}{\varepsilon^{2}df}+1,$$
where *α* = 0.05; *β* = 0.90; *ε* = 0.05; *δ*_1–*β*_ ≈ 27.939; and *df* = 24.

Participants were recruited via posters and take-home letters providing details about the study. Information about the program and how to participate in the study was also shared through common social networks (e.g., parent meetings). Interested parents were self-screened for study eligibility according to the distributed participant information sheet, and informed consent was obtained prior to data collection. A total of 870 parents were recruited at baseline. They all participated in the study at T2. We excluded parents who were not responsible for child feeding nor the primary caregivers, children with extreme age-standardized body mass index (BMI) *Z*-scores or nutrition-related diseases (e.g., eating disorders), and those with missing values > 10% in all included variables. Finally, 853 (98.05%) responses were included in this study.

We trained healthcare staff in the kindergartens to collect data at two time points with a 6-month interval. They explained the aim of the study to the participants to increase parental willingness to participate in the parent meeting. Parents were highly motivated to participate in this study due to the importance of feeding practices in parenting. The research team also provided detailed feedback if the participants had questions about the study. Ethical approval for the study was obtained from the Research Ethics Committee of Shanghai Jiao Tong University (approval number: SJUPN-201908). We obtained written informed consent from all participants at each time point.

### 2.2. Demographic and Socioeconomic Data

A self-reported questionnaire was used to obtain demographic and socioeconomic data, including the children’s age, sex, and duration of breastfeeding; caregivers’ role, age, weight, height, education level, and household annual income; the number of children; and the family structure. Parental weight status was classified as underweight (BMI < 18.5 kg/m^2^), normal weight (18.5 kg/m^2^ ≤ BMI < 24.0 kg/m^2^), or overweight or obese (BMI > 24.0 kg/m^2^) [30]. Child height (to the nearest 0.1 of a centimeter) and weight (to the nearest 0.1 of a kilogram) were obtained by trained kindergarten healthcare staff using standardized anthropometric equipment. Children were lightly clothed for all measurements, with shoes removed. According to the World Health Organization (WHO) guidelines, the children’s age-standardized BMI Z-scores were calculated, adjusting for age and gender, using the software WHO Anthro version 3.2.2, 2011 (for 2- to 5-year-old children) and WHO AnthroPlus version 2009 (for 5- to 6-year-old children). Child BMI *Z*-scores were categorized into four groups: underweight (*Z*-score < −2), normal weight (−2 ≤ *Z*-score < 1), overweight (1 ≤ *Z*-score < 2), and obese (*Z*-score ≥ 2) [31].

### 2.3. PFPs

The Chinese Preschoolers’ Caregivers’ Feeding Behavior Scale (CPCFBS) was used to evaluate PFPs [32]. The CPCFBS assesses two types of NFPs: four items of content-restricted feeding (i.e., restricted access to unhealthy food or opportunities to consume unhealthy food), three items of pressure to eat (i.e., insists, demands, or physical struggles with the child to get the child to eat more food), and three types of RFPs: four items of monitoring (i.e., the extent to which caregivers oversee their children’s eating), six items of encouraging healthy eating (i.e., the behavior of encouraging their children to eat more healthy food), and seven items of modeling (i.e., the modeling of healthy food choices to encourage children to adopt similar behaviors). Two items of use of food as a reward (i.e., use of desired food as a method to regulate child eating or behaviors) were assessed using the Chinese version of the Child Feeding Questionnaire (C-CFQ) [33]. Each item of the CPCFBS and C-CFQ was rated on a 5-point Likert scale. The response options for each item were “always”, “usually”, “sometimes”, “rarely”, and “never”. Each subscale was calculated by averaging the scores of all items in that subscale. The CPCFBS demonstrated good internal consistency reliability at two time points (Cronbach’s α at T1 = 0.725–0.87 and at T2 = 0.740–0.895), and the internal consistency of the food-as-a-reward subscale of the C-CFQ was moderate (Cronbach’s α at T1= 0.585 and at T2 = 0.657).

### 2.4. CEBs

Five types of CEBs were assessed by the Chinese Preschoolers’ Eating Behavior Questionnaire (CPEBQ) [34], including five items of food fussiness (i.e., reluctance to try new food or eating limited food), six items of food responsiveness (i.e., children’s desire to eat food when they see or smell food or are supplied with food), five items of satiety responsiveness (i.e., the limited amount of food the child eats in a meal), four items of unhealthy eating habits (i.e., the behaviors of child chewing, swallowing, spitting out or throwing, etc.), and five items of initiative eating (i.e., the ability of the child to eat independently). Each item was rated on a 5-point Likert scale. Each subscale was calculated by averaging the scores of all items in that subscale. The CPEBQ had good internal consistency reliability at two time points (Cronbach’s α at T1 = 0.703–0.773 and at T2 = 0.719–0.785).

### 2.5. Covariates

In addition to demographic and sociological factors, several previously identified potential confounders below were included in the study.

#### 2.5.1. Parental Perception of Child Weight and Concern about Weight

The parental perception of child weight was assessed by asking, “How would you describe your child’s weight?”. Parental concern about child overweight and concern about child underweight were assessed separately by asking, “How concerned are you about your child becoming or staying overweight in the future”? and “How concerned are you about your child becoming or staying underweight in the future”? Each item was rated on a 5-point Likert scale. These items have been used and validated in previous studies [35].

#### 2.5.2. Child Temperament

The Chinese version of the Children’s Behavior Questionnaire (C-CBQ) was used to assess Inhibitory Control and Anger/Frustration [36]. Inhibitory Control (six items) refers to the capacity to plan and control inappropriate approach responses under instructions or in novel or uncertain situations [36]. Anger/Frustration (six items) refers to the extent of the negative emotional effect related to the interruption of ongoing tasks or goal blocking [36]. Parents were asked to rate their child on each item using a 7-point Likert scale. Each subscale was calculated by averaging the scores of all items in that subscale. The Inhibitory Control and Anger/Frustration scale showed moderate and good internal consistency reliability at T1 in our study (Cronbach’s α = 0.609 and 0.757, respectively).

#### 2.5.3. Parental Depression

Parental depression was evaluated using the Chinese version of the 10-item Center for Epidemiologic Studies Depression Scale (CES-D) [37]. Each item ranged from 0 to 3, and a higher numerical response represented a greater expression of depressive symptoms. The total score was obtained by summing up the scores of the ten items. The Cronbach’s α in our study at T1 was 0.721.

## 3. Statistical Analysis

Data were checked for normality and missing values. Descriptive statistics were used to describe the participants’ characteristics. Independent *t*-tests and one-way analysis of variance were performed to compare demographic and sociological differences in the mean scores of PFPs and CEBs. We examined the relationships between continuous variables using Pearson’s correlation analysis. All of the studied variables were approximately normally distributed, with skewness and kurtosis below 2.0. The continuity of PFPs and CEBs was tested using paired *t*-tests to see if the mean score differed significantly between the two time points. Pearson correlations were used to examine the stability of PFPs and CEBs across two time points. These methods have been used in previous studies [13,38]. Parameters were estimated using cross-lagged analyses via maximum likelihood estimation. In the cross-lagged models, we adjusted for covariates significantly associated with CEBs at T1 (*p* < 0.05). The goodness of fit of the models was assessed using the following indices and cut-offs: χ^2^/*df* about 3 or smaller, goodness-of-fit index (GFI) and comparative fit index (CFI) > 0.90, root mean square residual (RMR) < 0.08, and root mean square error of approximation (RMSEA) < 0.06 [39,40], which was used in some studies [35]. Multiple imputation by chained equations (MICE) was adopted to impute missing values because over 5% (6.2%, *n* = 53) of the participants had missing data on parental self-reported feeding practices and CEBs. The MICE procedure used all available variables with complete information in the study [41]. There were no significant differences in demographic and socioeconomic characteristics between those with missing data and those without. Statistical significance was set at *p* < 0.05 (two-sided). SPSS statistics 27.0 (IBM Crop, Armonk, NY, USA) and Mplus 7.4 (Muthén & Muthén, Los Angeles, CA, USA) were used for data coding, cleaning, and analysis.

## 4. Results

### 4.1. Demographic and Socioeconomic Characteristics of the Participants at Baseline

The mean ages of the caregivers and children were 35.21 years (standard deviation (SD) = 4.22 years) and 4.39 years (SD = 0.72 years), respectively. A total of 700 caregivers (82.1%) were mothers, and 452 (53.0%) children were boys. One hundred and thirty-eight preschool children were overweight (*n* = 109, 12.8%) or obese (*n* = 29, 3.4%). The education level of the caregivers was high, with 90.8% finishing college or higher education. About sixty-two percent of the families had only one child, and more than half (58.3%) had an annual income below CNY 300,000 (Table 1).

Appendix A shows the bivariate analyses between the demographic characteristics, PFPs, and CEBs at T1. Some demographic variables were statistically significantly related to CEBs at T1 (Appendix A) and were added to the corresponding cross-lagged model as covariates.

### 4.2. Stability and Continuity of PFPs and CEBs

PFPs (*r* = 0.419–0.576, *p* < 0.001) and CEBs (*r* = 0.536–0.690, *p* < 0.001) showed moderate to high stability across successive 6-month periods (Table 2).

The mean scores of PFPs were higher than the midpoint of the Likert scale at both T1 and T2 (Table 2). The scale of parental modeling had the highest mean score at T1 (4.116 ± 0.567), encouragement of healthy eating had the highest score at T2 (4.135 ± 0.529), and the lowest ones were for pressure to eat (3.217 ± 0.849 at T1 and 3.291 ± 0.811 at T2). Compared with T1, parental content-restricted feeding (3.451 vs. 3.724, *p* < 0.001) and the use of food as a reward (3.271 vs. 3.348, *p* < 0.001) were statistically significantly higher at T2.

The mean scores of CEBs (except for initiative eating) were close to or lower than the midpoint of the Likert scale at T1 and T2 (Table 2). Children’s initiative eating had the highest mean scores at both T1 (3.461 ± 0.673) and T2 (3.545 ± 0.653). The scale of child food responsiveness had the lowest mean scores at T1 (2.445 ± 0.568), and unhealthy eating habits had the lowest mean scores at T2 (2.422 ± 0.787). The results showed that the mean score of satiety responsiveness was statistically significantly higher at T2 than at T1 (2.778 vs. 2.648, *p* < 0.001).

### 4.3. Correlations between PFPs and CEBs

Appendix A shows the bivariate correlations between PFPs and CEBs. Their statistically significant cross-sectional and longitudinal correlations show low to moderate effects.

### 4.4. Bidirectional Associations of PFPs with CEBs

In total, sixty longitudinal cross-lagged paths were tested in our study, of which eighteen were found to be prospectively significant. The results from the structural equation models revealed a good fit of the data (Appendix A) [39,40].

The cross-lagged paths between RFPs and CEBs are illustrated in Figure 1. Only one model showed that both lagged effects were significant, with the path from encouragement of healthy eating to food fussiness (standardized *β* = −0.077, standard error (SE) = 0.028, *p* < 0.001) being slightly stronger than the reversed path (standardized *β* = −0.065, SE = 0.028, *p* < 0.01). In addition, four parent-driven associations and one child-driven association were reported. In particular, the results showed that parental encouragement of healthy eating negatively predicted children’s unhealthy eating habits (standardized *β* = −0.051, SE = 0.025, *p* < 0.05), modeling negatively predicted food fussiness (standardized *β* = −0.068, SE 0.027, *p* < 0.05) and unhealthy eating habits (standardized *β* = −0.061, SE = 0.025, *p* < 0.05), and monitoring negatively predicted children’s unhealthy eating habits (standardized *β* = −0.066, SE = 0.025, *p* < 0.01). The only child-driven association is that child initiative eating positively predicted parental encouragement of healthy eating (standardized *β* = 0.064, SE = 0.028, *p* < 0.05).

Figure 2 shows the cross-lagged paths between NFPs and CEBs. One statistically significant bidirectional association was observed. The standardized *β* for the path from content-restricted feeding to food fussiness was −0.071 (SE = 0.27, *p* < 0.001), and that for the reversed path was −0.071 (SE = 0.031, *p* < 0.01). Furthermore, one parent-driven association and eight child-driven associations were identified. Specifically, child food fussiness positively predicted parental use of food as a reward (standardized *β* = 0.109, SE = 0.031, *p* < 0.001); satiety responsiveness negatively predicted content-restricted feeding (standardized *β* = −0.063, SE = 0.031, *p* < 0.05) and positively predicted pressure to eat (standardized *β* = 0.057, SE = 0.029, *p* < 0.05) and the use of food as a reward (standardized *β* = 0.083, SE = 0.031, *p* < 0.01); and unhealthy eating habits positively predicted pressure to eat (standardized *β* = 0.098, SE = 0.030, *p* < 0.01) and the use of food as a reward (standardized *β* = 0.086, SE = 0.031, *p* < 0.01). However, child initiative eating negatively predicted pressure to eat (standardized *β* = −0.071, SE = 0.029, *p* < 0.05) and the use of food as a reward (standardized *β* = −0.067, SE = 0.031, *p* < 0.05). The only parent-driven association is that parental content-restricted feeding negatively predicted children’s unhealthy eating habits (standardized *β* = −0.083, SE = 0.025, *p* < 0.05).

## 5. Discussion

In this longitudinal study, we used cross-lagged analyses to examine the bidirectional relationships between six types of PFPs and five types of CEBs. To the best of our knowledge, this study is the first prospective study examining their associations in a Chinese sample.

The cross-lagged analyses based on structural equation modeling revealed eighteen statistically significant paths from all sixty longitudinal paths between PFPs and CEBs. There were two bidirectional associations, five parent-driven associations, and nine child-driven associations. Specifically, the current results showed that parental encouragement of healthy eating and content-restricted feeding had bidirectional associations with child food fussiness. More parental use of encouragement of healthy eating and content-restricted feeding predicted lower child food fussiness reported by parents six months later, and vice versa, suggesting that some specific feeding practices and eating behaviors may have substantial interactions over time. Food fussiness is one of the most common CEBs [42], which may result in caregivers’ attention and the adoption of specific feeding practices (e.g., encouragement of healthy eating and content-restricted feeding) in response to it. Similarly, these two feeding practices were shown to be often used in our sample and may have resulted in significant changes in child food fussiness after six months. Our findings are in line with some theories (e.g., Ecological System Theory [19]) and research in developmental psychology suggesting that PFPs and CEBs are bidirectionally associated in the family environment [19]. Some empirical evidence supported their bidirectional associations [21,22,23]. However, Mallan et al. [43] reported no significant longitudinal associations between overt maternal restriction and child food fussiness at the 1.3-year follow-up and 1.7-year follow-up. An important factor may be differences in sample characteristics (e.g., child age). Older children may be more aware of PFPs (e.g., restrictive feeding) than younger children and thus more likely to respond to feeding practices (e.g., eating in the absence of hunger) [13,44]. In addition, different confounders that have been controlled for among existing studies may result in inconsistencies between the findings [45,46]. Finally, some studies may be underpowered to detect a significant association. An adequately powered sample is further needed, considering that numerous factors are associated with these behaviors [25,47].

Our study showed that most of the significant paths between RFPs and CEBs were parent-driven associations. That is, parental modeling, the encouragement of healthy eating, and monitoring negatively predicted children’s unhealthy eating habits, and modeling negatively predicted child food fussiness six months later. Our findings highlight the key roles of RFPs in fostering children’s good eating behaviors, such as helping children accept a variety of foods, similar to previous studies [12,48]. Parents may be more likely to structure the family eating environment by providing healthy food, modeling, or monitoring if they realize the negative effects of non-responsive feeding practices (e.g., pressure to eat) on their children’s eating, weight, and nutrition [7]. In contrast to more overt forms of control or pressure (e.g., forced feeding), a responsive feeding strategy may allow children to develop self-regulatory skills, which can help them develop healthy eating habits [49]. It is worth noting that parents used RFPs more frequently than NFPs in our sample, which may increase the likelihood of improving child eating habits. We also found one child-driven association between RFPs and CEBs: child initiative eating positively predicted parental encouragement of healthy eating. This eating behavior had the highest mean score, indicating that it frequently occurred in these children, and parents may be more likely to apply feeding practices appropriately in response to their children’s food interests (e.g., encouragement of healthy eating). This finding suggests that children’s good performance in eating might encourage positive responses from the parents (e.g., adopting RFPs), in line with previous evidence [15].

Meanwhile, our study revealed eight child-driven associations and one parent-driven association between NFPs and CEBs. In particular, after six months, satiety responsiveness positively predicted the use of food as a reward and pressure to eat and negatively predicted content-restricted feeding; children’s unhealthy eating habits positively predicted the use of food as a reward and pressure to eat; food fussiness positively predicted the parental use of food as a reward; and initiative eating negatively predicted pressure to eat and the use of food as a reward. Consistent with recent findings [21,23], our findings indicate that parents’ adoption of NFPs might be in response to their perceptions of their children’s problematic eating behaviors. Parents may see coercive control feeding practices as a straightforward method for controlling their children’s specific eating behaviors, such as limiting their unhealthy food intake [7]. The results also showed that parental concern about both child overweight and underweight had close relationships with most eating behaviors (Appendix A). Parents who were more concerned about children overweight or underweight might have paid greater attention to their children’s eating behaviors. Simultaneously, some eating behaviors (e.g., satiety responsiveness and unhealthy eating habits) may raise parental worries about child nutrition and weight status [15]. As a result, they may use inappropriate feeding practices, such as pressure to eat [47]. Similarly, we found that satiety responsiveness, food fussiness, and unhealthy eating habits positively predicted the use of food as a reward. Caregivers from developing countries (e.g., China) or low socioeconomic settings prefer to express love and care through food and use “no favorite food” as a method of punishment [26,35]. They may not consider being overweight as a health problem [26,27], as supported by our study, where parents expressed more concerns about their children being underweight than overweight (2.24 vs. 1.78) (Appendix A) and showed a low level of concern about child overweight when compared to caregivers in other countries [47]. Chinese caregivers may become more concerned about their children’s health issues if their children show less appetite for food (e.g., satiety responsiveness) because they believe that the more children eat, the better they will be [26,27]. In this case, parents may be more likely to use food as a reward or an educational tool to make their children eat more. Furthermore, Chinese parents are likely to notice some of their children’s eating behaviors, such as food fussiness, emotional eating, and satiety responsiveness, and thus, are inclined to remedy them using NFPs. For example, a qualitative study (*n* = 22) conducted among Chinese mothers in the US revealed that mothers tended to adopt forced feeding when their children could not finish a specified amount of the meal, or they found that their children did not eat enough when compared with other children [26]. This may be due to feeding stress and concern in response to some dietary issues with their children [26,50]. In contrast, our study indicated that greater child initiative eating predicted lower parental use of food as a reward and pressure to eat. This indicates that a high frequency of good performance in child eating may help parents reduce the use of inappropriate feeding practices.

We also found one parent-driven association between NFPs and CEBs: parental content-restricted feeding negatively predicted children’s unhealthy eating habits. This finding differs from those of previous studies in developed countries [43,46]. This inconsistency may be due to the complex nature of the parental control of child eating, which might not be captured by the current measurement [32,44]. For example, a study conducted in Australia with 252 mothers of children aged 3–11 years old reported that more restricted feeding strategies predicted higher unhealthy snack intake, while higher covert control predicted lower unhealthy snack intake with a 3-year follow-up [48]. Parental coercive control feeding might inhibit children from learning to self-regulate their eating, resulting in a greater desire for restricted food and unhealthy food intake [49]. The obvious type of control in the CPCFBS is “overt”, which entails restricting children’s unhealthy food in a way that children can notice. The items might reflect caregivers limiting what their children eat by controlling their food environment by avoiding places that sell unhealthy food and only permitting healthy choices. The other type of control is described as “covert”, as it remains undetected by children, which may have no direct effects on limiting the child’s self-regulation of eating and hence not result in their unhealthy food intake [7].

Additionally, the cross-sectional and longitudinal correlations between feeding practices and CEBs were not strong, and more than two-thirds of longitudinal associations showed no statistical significance. In our study, some covariates (e.g., child temperament and parental concern about child weight) had close links to CEBs, which may weaken the relationships between PFPs and CEBs. For example, a quasi-experimental study (*n* = 37) found that preschool children with a lower level of inhibition control showed a greater increase in restricted food intake in response to parental restrictive feeding. In contrast, children with a higher level of inhibition control showed no significant changes [49]. It seems that the children’s temperaments played an important role in these relationships, which might influence the estimates of paths if not added as covariates. In addition, we used cross-lagged analyses to explore the bidirectional associations between specific feeding practices and eating behaviors, whereas CEBs were influenced by a combination of food-related parenting practices in real time [15,51]. This may be one of the reasons that we found non-significant associations in most models. Future studies should consider the combined effects of PFPs and CEBs on each other.

A major strength of this study is the collective thoughts on the prospective associations between different kinds of feeding practices and eating behaviors on the basis of cross-lagged models using SEM while controlling for several covariates. However, there are several limitations to this study. Convenience sampling has limited generalizability. Most parents recruited in this study had a relatively high level of education and therefore cannot represent those from rural areas with lower education. Random samples from multiple sites are further required. Another limitation is that recall bias cannot be eliminated from self-reported data. We conducted cross-lagged analyses at two time points with only a 6-month interval and thus could not explore the trajectory of PFPs and CEBs over a long time span. Furthermore, we examined the bidirectional associations without considering the effects of a combination of feeding practices since there is no appropriate assessment tool to measure two domains of feeding practices (NFPs and RFPs) in China. Therefore, there is a need to develop such comprehensive assessment tools in the future.

## 6. Conclusions

The cross-lagged models analyzed in this study helped elucidate the bidirectional interrelationships between PFPs and CEBs. Statistically significant bidirectional, parent-driven, and child-driven associations were observed between PFPs and CEBs. The findings indicate that RFPs predicted a decrease in children’s food fussiness and unhealthy eating habits, while NFPs were triggered by children’s food fussiness, satiety responsiveness, and unhealthy eating habits. These insights suggest that interventions should be developed to assist parents in recognizing, understanding, and responding appropriately to their CEBs. Such interventions may prove beneficial for parents in fostering and maintaining healthy eating habits in their children.

## Figures and Tables

**Figure 1 nutrients-16-00044-f001:**
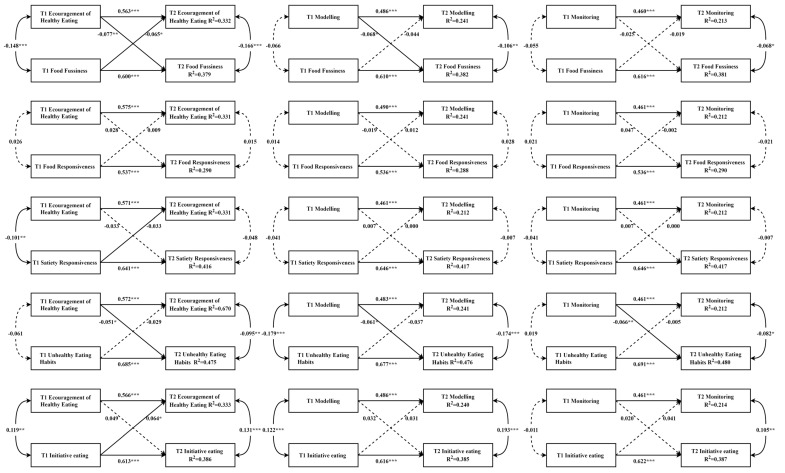
Cross-lagged models of associations between RFPs and CEBs. Notes. All models were adjusted for covariates. Arrows represent directions. Significant paths are shown with solid lines, and non-significant paths are shown with dashed lines. * *p* < 0.05, ** *p* < 0.01, *** *p* < 0.001.

**Figure 2 nutrients-16-00044-f002:**
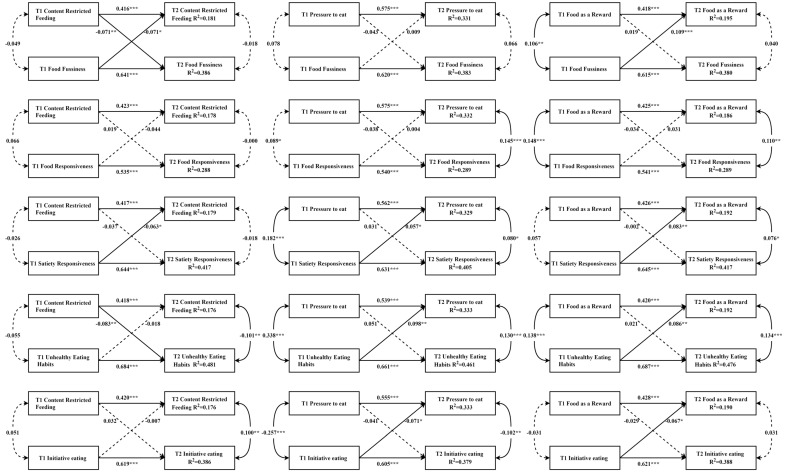
Cross-lagged models of associations between NFPs and CEBs. Notes. All models were adjusted for covariates. Arrows represent directions. Significant paths are shown with solid lines, and non-significant paths are shown with dashed lines. * *p* < 0.05, ** *p* < 0.01, *** *p* < 0.001.

**Table 1 nutrients-16-00044-t001:** Demographic characteristics of the participants at baseline (*n* = 853).

Variables	Mean (SD)/N (%)
Child age (years)	4.39 (0.72)
Child sex	
Boys	452 (53.0)
Girls	401 (47.0)
Child weight (BMI *Z*-score)	
Underweight: BMI *Z*-score < −2	25 (2.9)
Normal weight: −2 ≤ BMI *Z*-score < 1	690 (80.9)
Overweight: 1 ≤ BMI *Z*-score < 2	109 (12.8)
Obesity: BMI *Z*-score ≥ 2	29 (3.4)
Children’s temperament	
Inhibition control (range of possible scores: 1–7)	5.259 (0.78)
Emotionality (range of possible scores: 1–7)	4.045 (0.97)
Duration of breastfeeding	
0–6 months	239 (28.0)
6–12 months	415 (48.7)
More than 12 months	199 (23.3)
The role of caregivers	
Mothers	700 (82.1)
Fathers	153 (17.9)
Parental age	35.21 (4.22)
Parental weight status	
Underweight: BMI < 18.5 kg/m^2^	68 (8.0)
Normal weight: 18.5 kg/m^2^ ≤ BMI < 24.0 kg/m^2^	567 (66.7)
Overweight or obesity: BMI ≥ 24.0 kg/m^2^	215 (25.3)
Parental education level	
Senior high school or below	79 (9.2)
College or higher	774 (90.8)
Parental perception of child weight (range of possible scores: 1–5)	2.81 (0.70)
Parental concern about child overweight (range of possible scores: 1–5)	1.78 (0.97)
Parental concern about child underweight (range of possible scores: 1–5)	2.24 (1.08)
Number of children	
One	532 (62.4)
Two or more	321 (37.6)
Family structure	
Living with parents and grandparents	454 (53.2)
Living with parents	399 (46.8)
Household income/year	
Above average (>CNY 300,000)	350 (41.7)
Below average (≤CNY 300,000)	489 (58.3)
Parental depression (range of possible scores: 0–30)	4.14 (3.39)

**Table 2 nutrients-16-00044-t002:** Stability and continuity of PFPs and CEBs (*n* = 853).

Measure	T1 Mean ± SD	T2 Mean ± SD	Pearson’s r	t	*p*
PFPs					
Encouragement of healthy eating	4.111 ± 0.510	4.135 ± 0.529	0.575 ***	−0.924	0.356
Modeling	4.116 ± 0.567	4.128 ± 0.602	0.490 ***	−0.437	0.662
Monitoring	4.056 ± 0.803	4.075 ± 0.737	0.461 ***	−0.509	0.610
Content-restricted feeding	3.451 ± 0.801	3.724 ± 0.749	0.419 ***	−7.260	<0.001
Pressure to eat	3.217 ± 0.849	3.291 ± 0.811	0.576 ***	−1.828	0.068
Use of food as a reward	3.271 ± 0.752	3.348 ± 0.782	0.430 ***	−2.094	0.036
CEBs					
Food fussiness	3.005 ± 0.648	2.958 ± 0.658	0.617 ***	1.704	0.138
Food responsiveness	2.445 ± 0.568	2.443 ± 0.584	0.536 ***	0.043	0.965
Satiety responsiveness	2.648 ± 0.682	2.778 ± 0.665	0.645 ***	−3.965	<0.001
Unhealthy eating habits	2.490 ± 0.811	2.422 ± 0.787	0.690 ***	1.779	0.075
Initiative eating	3.461 ± 0.673	3.545 ± 0.653	0.622 ***	−2.610	0.009

Notes. *** *p* < 0.001.

## Data Availability

The data presented in this study are available on request from the corresponding author. The data are not publicly available due to containing information that could compromise the participants’ privacy.

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
