# Peer review of "Bidirectional Associations between Parental Feeding Practices and Child Eating Behaviors in a Chinese Sample"

_nutrients, 2023, doi:10.3390/nu16010044_

Round 1

Reviewer 1 Report

Comments and Suggestions for Authors

This is a very interesting and timely paper looking at the bidirectional relationships between parental feeding practices (PFP’s) and child eating behaviours (CEB’s). Childhood obesity is rising in China and there is a real need to examine the influence of parental feeding practices on their children’s eating behaviours.

Abstract

Line 3 – recommend rephrasing ‘were equivocal’ to ‘remain equivocal’

Introduction

Page 2 Line 5 recommend rephrasing ‘They were classified into……’ to PFP have been classified into two food parenting constructs….’

Page 2 last para Line 6 When writing about obesity in the literature it is usual now to write of people with obesity as opposed to obese people. Recommend rephrasing ‘children who are overweight or obesity’ to ‘children with overweight or obesity’ .

Methods

These could have been more detailed to give the reader more clarity on how the study data was collected.

This was – as stated by the authors a convenience sample. Little/No detail is provided on how participants were actually recruited and by whom.  More transparency in the methods would be useful to enable clearer interpretation of the findings from the study.

In relation to the 6 month follow-up, was this an exact date for each individual or was there variability around this? Mean 6 months +/_ SD? The authors mention that this is a time interval that has been used in previous studies, however these two previous studies were on toddlers (somewhat younger than this sample of children).The time interval is short – was there a rationale for this? If a prospective study, could the authors consider looking at these relationships between PFP’s and CEB’s at other time points in addition to 6 months such as at 12 months, 18 months and 2 years?

The authors mention that the children’s height and weight (for calculating BMI status) were recorded by health care teachers? Were they trained? Was there a protocol for them? What instruments did they use for taking these measurements. These should be described in the methods.

Pearson correlations were used to examine the stability of PFP’s and CEB’s across the two time points. By stability do the authors mean agreement? In which case Bland and Altman plots may be more appropriate? In examining the differences (or lack of – i.e. stability) in these measures, it may surely be possible that over the 6 months there may have been actual changes in PFP’s or in CEB’s. For example, food fussiness peaks in toddlers and then starts to reduce.

Results

These were verbally described by the authors in the present tense, it is more usual to describe the results in the past tense. Use of ‘were’ instead of ‘are’.

Discussion

The authors have done a commendable job in discussing their findings in terms of the literature and in cases where their results may have differed from previous studies they provide possible reasons to explain such differences. Having said this, I am not sure that the explanation provided in Discussion Line 24 of discussion section where this study differed to that of Mallan et al. (ref 42) due to their study having longer time intervals and therefore as a result missing associations. The only way to determine this would be to look at this at several time points to see what associations look like at shorter and longer time intervals and the extent to which these associations strengthen or weaken.

The authors mention how Chinese perceive chubby even overweight children as healthy which is quite different to many other countries (developed). The authors make this point earlier in the paper and imply their country is not a developed country. How do the authors classify China – as a developed, transition or developing country?  The cultural differences in weight perception, its healthiness and desirability would be worthy of some discussion as to how this may influence associations observed in this study and in particular how it may result in differences to other study findings.

Discussion Line 41 needs to be rephrased for clarity ‘they may decrease the likelihood of making force towards child eating’ as not sure what the authors mean here.

The authors doing a good job in providing the limitations of the study, they could also give some of the strengths too such as adequate power to pick up associations. The use of SEM for examining these associations.

Comments on the Quality of English Language

The paper is well written with just a few minor editorial details and clarification of sense. These have been noted above.

Author Response

Thank you very much for your constructive feedback. Please find the attached Response Letter.

Reviewer 2 Report

Comments and Suggestions for Authors

The work sent for review is a well-prepared research report. The literature review provides a good introduction to the content of the study. The purpose of the study is well specified and described. The study addresses the important issue of the formation of eating habits in children and its relationship to parental factors. The study was performed in a large group of parents and children. The method of selecting subjects for the study was well described. The research methods and methods of statistical processing of the results were correctly selected and described. The results were correctly presented and properly described. The discussion was conducted in a clear manner, with a critical interpretation of one's own results and with reference to the results of other authors. The conclusions are adequate for the results obtained. The study provides important information on the relationship between partntal feeding ptractises and child etating behaviores in a Chinese sample. The authors also took into account the existence of limitations for the study and for the interpretation of the obtained results.

Author Response

Thank you very much for the review of our manuscript. We are grateful for your time and your kind words on this work.